# Strong Linear Correlation between CH_3_NH_2_ Molecular Defect and THz-Wave Absorption in CH_3_NH_3_PbI_3_ Hybrid Perovskite Thin Film

**DOI:** 10.3390/nano10040721

**Published:** 2020-04-10

**Authors:** Inhee Maeng, Asuka Matsuyama, Jung-Ho Yun, Shenghao Wang, Chul Kang, Chul-Sik Kee, Masakazu Nakamura, Min-Cherl Jung

**Affiliations:** 1Advanced Photonics Research Institute, Gwangju Institute of Science and Technology, Gwangju 61005, Korea; inheem@yuhs.ac (I.M.); iron74@gist.ac.kr (C.K.); cskee@gist.ac.kr (C.-S.K.); 2Division of Materials Science, Nara Institute of Science and Technology, Nara 630-0192, Japan; matsuyama.asuka.lr5@ms.naist.jp (A.M.); mnakamura@ms.naist.jp (M.N.); 3School of Chemical Engineering and Australian Institute for Bioengineering and Nanotechnology (AIBN), University of Queensland, Brisbane QLD 4027, Australia; j.yun1@uq.edu.au; 4Materials Genome Institute, Shanghai University, Shanghai 200444, China; shenghaowang@shu.edu.cn or; 5Division of Materials Science, Faculty of Pure and Applied Sciences, University of Tsukuba, Ibaraki 305-8577, Japan

**Keywords:** CH_3_NH_2_, THz oscillation strength, MAPbI_3_

## Abstract

To control the density of a CH_3_NH_2_ molecular defect, which strongly contributed to a significant THz-wave absorption property in the CH_3_NH_3_PbI_3_ hybrid perovskite thin film formed by the sequential vacuum evaporation method, we performed post-annealing processes with various temperatures and times. In the thin film after post-annealing at 110 °C for 45 min, the density of the CH_3_NH_2_ molecular defect was minimized, and CH_3_NH_3_I and PbI_2_ disappeared in the thin film after the post-annealing process at 150 °C for 30 min. However, the density of the CH_3_NH_2_ molecular defect increased. Moreover, the THz-wave absorption property for each thin film was obtained using a THz time-domain spectroscopy to understand the correlation between the density of a molecular defect and the THz-wave oscillation strength at 1.6 THz, which originated in the molecular defect-incorporated hybrid perovskite structure. There is a strong linear correlation between the oscillator strength of a significant THz-wave absorption at 1.6 THz and the CH_3_NH_2_ molecular defect density.

## 1. Introduction

Recently, the organic–inorganic hybrid perovskite (OHP) material ABX_3_ (A = Organic cation: CH_3_NH_3_^+^/NH_2_CH=NH_2_^+^, B = Metal cation: Pb/Sn, and X = Halide anion: Cl/Br/I) has been identified as a potential application material for solar cell, field-effect transistor, and light-emitting diode applications [1,2,3,4,5,6,7]. In the last decade, many works have been reported in OHP material research [8]. Even nano-scaled formation research using OHP material has just begun, and its single crystal formation with nano-size and physical properties, such as bandgap engineering, have been reported [9,10,11]. New applications employing these OHP materials are expected to be further expanded, due to several unique physical properties that are still unclear. In our previous research, we found several significant THz-wave absorptions in CH_3_NH_3_PbI_3_ (MAPbI_3_) and HC(NH_2_)_2_PbI_3_ (FAPbI_3_) that originated in a kind of defect structure [12,13,14]. Particularly, we found that the origin of absorptance in MAPbI_3_ was the CH_3_NH_2_ molecular defect-incorporated perovskite structure (significant I–Pb–I vibration mode) [13]. This finding shows an interesting possibility for a new application, such as THz-wave modulation, sensing, and imaging devices instead of high purified GaAs formed at low temperature with an expensive unit price and ultra-high vacuum-based fabrication [15,16]. This means that we need to find and improve on a controllable physical property, such as the strength control of an I–Pb–I vibration mode.

However, another problem arises, such as the material stability because there is no post-annealing process to keep the presence of a molecular defect [13,14,17,18,19]. Generally, the post-annealing process can induce more stable OHP with large grains and clear stoichiometry. On the other hand, we can guess easily that the neutral molecular defect such as CH_3_NH_2_ will disappear during post-annealing. In short, it is required to find an optimized condition to (1) ensure material stability in the OHP thin film and (2) maintain enough density of the CH_3_NH_2_ molecular defect for the significant vibration mode.

In this article, MAPbI_3_ thin films fabricated with the sequential vacuum evaporation (SVE) method [12,13,20,21] are performed using the post-annealing process with various temperatures and times. We then investigate their atomic structures, defects, and chemical states to find an optimized state with the control of defect density and the improvement of stability. Finally, we look for a linear correlation between the density of CH_3_NH_2_ molecular defect and the THz-wave oscillator strength to lead to the key possibility of new applications, such as THz-wave modulation, sensing, and imaging devices.

## 2. Materials and Methods 

OHP thin films were fabricated using the SVE method in the customized vacuum chamber [13,21]. A silicon substrate (*n*-type doped Si(100)) was cleaned by sonication in acetone for 10 min and then rinsed in heated acetone for 1 min. After this, a UV-ozone treatment was performed for 30 min before loading it into a vacuum chamber. In addition, the Al_2_O_3_ substrates (made by Hi-Solar Co., Ltd., Gwangju, Korea) were used for the measurement of THz time-domain spectroscopy (THz-TDS). The surface orientation is C-plane(0001) with an off-angle of 0.2° ± 0.05°. The thickness and roughness are 430 ± 25 μm and Ra ≤ 0.3 nm, respectively. The cleaning method is the same. The base pressure of the chamber installed with the thickness monitor sensor was 8.0 × 10^−3^ Pa. A Lead(II) iodide (PbI_2_, 99% purity, Sigma-Aldrich, St. Louis, MO, USA) was evaporated with a deposition rate of 10 Å/s onto the substrates at room temperature to form the film with a 100 nm thickness (Figure 1). Continuously, a methylammonium iodide (CH_3_NH_3_I, 98% purity, Sigma-Aldrich, St. Louis, MO, USA) was evaporated with a deposition rate of 2 Å/s and a 280 nm thickness onto the formed PbI_2_ thin film. Finally, we obtained the MAPbI_3_ thin films with a thickness of 300 nm [21] (Figure 1). To see the temperature dependence of MAPbI_3_, we performed the annealing processes at a temperature of 110 °C for 45 min (the typical annealing condition for solar cell application) and 150 °C for 10 and 30 min (to avoid any dramatical material degradation) [21]. Over a temperature of 150 °C, the depletion/degradation processes of MAPbI_3_ were so fast that it was very difficult to maintain a hybrid perovskite structure. To characterize all of the formed and treated thin films, we performed scanning electron microscopy (SEM), X-ray diffraction (XRD), and high-resolution X-ray photoelectron spectroscopy (XPS). The used SEM system is the HITACHI SU9000 model (Tokyo, Japan) with an acceleration voltage of 5.0 kV and an emission current of 10 μA. The model of XRD with a Cu*K*_α_ source is RINT-TTRIII/NM made by Rigaku (Tokyo, Japan). We used the VersaProbe II (Chigasaki, Japan) with a monochromated Al*K*_α_ (ULVAC-PHI, (Chigasaki, Japan) for all XPS measurements and obtained the C 1*s*, N 1*s*, Pb 4*f*, and I 4*d* core-level spectra. In all samples, no traces of O_1s_ core-level spectra were observed. The binding energies were calibrated with reference to the Au 4*f*_7/2_ level (84.0 eV) [22]. The THz spectra were measured with a standard THz-TDS transmission setup based on a Ti:Sapphire regenerative amplifier system, which has 35 *fs* pulses at an 800 nm wavelength and a repetition rate of 1 kHz. We used a 2 mm ZnTe(110) crystal for generation and detection. We measured the THz time-domain signal in the spectral range from 0.5 up to 2.5 THz. We measured the THz-TDS spectra of the formed and treated thin films on the Al_2_O_3_ substrate, and the THz conductivity was obtained from a thin film approximation equation [23].

## 3. Results and Discussion

In the no-annealed sample (A), we can observe a tiny grain structure (Figure 2a). However, its grain boundary is not clear. After annealing at 110 °C for 45 min (B), which is a typical annealing condition for a solution-processed thin film in solar cell application, the grains appear clearly with a grain size of 70–120 nm (Figure 2b). While increasing the annealing temperature at 150 °C for 10 min, the grain size increased over 200 nm (Figure 2c). By increasing the annealing time for 30 min at 150 °C, however, many small structures occurred at the grain boundary, and it looks like a surface degradation (Figure 2d). From these SEM results, we can agree that (1) the annealing process with a high temperature can increase the grain size, and (2) the increased annealing time can cause a degradation at the grain boundary [13,14,20,21]. 

To understand the structural formation, we performed the XRD experiment for all treated samples (Figure 2e). In all samples, the typical atomic structures of MAPbI_3_ were observed, such as (110), (112), (202), (220), (222), (204), and (312) in the range of 5–40° [21]. In the A sample, we can observe the trace of CH_3_NH_3_I (MAI), PbI_2_, and intermediate phases (the CH_3_NH_2_ incorporated hybrid perovskite structure) [21]. In the B sample, however, these structures have completely disappeared. We confirmed that the typical annealing condition for a solar cell application could remove the MAI and PbI_2_ phases formed by the SVE method [13,21]. When we increased the annealing temperature from 100 to 150 °C (the sample C), the MAI, PbI_2_, and the intermediate phases were still alive. When increasing the annealing time from 10 to 30 min at 150 °C (the D sample), only the intermediate phase, which originated in the CH_3_NH_2_ molecular defect-incorporated perovskite structure, has remained mainly with the hybrid perovskite phase, and the MAI and PbI_2_ phases have wholly disappeared [21].

To see changes of chemical states in all treated samples, we performed a high-resolution XPS experiment. The C 1*s*, N 1*s*, Pb 4*f*, and I_4d_ core-level spectra were obtained for each sample (Figure 3). First of all, we can notice that the chemical states of N 1*s* (402.7 eV at the maximum intensity peak position) and I 4*d* (I 4*d*_5/2_ = 619.5 eV) almost did not change on any annealing conditions [13,21] (Figure 3b,d). The A sample was observed consistently with our previous reports [20]. The typical chemical states of CH_3_NH_3_^+^ cation (C 1*s*—287.0 eV and N 1*s*—402.7 eV) and the CH_3_NH_2_ molecular defect (C 1*s*—286.0 eV and N 1*s*—403.3 eV) were observed in the C 1*s* and N_1s_ core-level spectra [13,21] (Figure 3a,b). The peak intensity of the CH_3_NH_2_ molecular defect decreased after annealing, and the Pb^0+^ chemical state (Pb metal) appeared at the 150 °C annealing temperature [13,21] (Figure 3a,c). This means that the hybrid perovskite structure on the surface was broken at the high-temperature annealing process. Moreover, the molecular defect and I-related elements were depleted from the surface, and only the Pb metal element remained on the surface.

To see the detailed behavior of the CH_3_NH_2_ molecular defect processed under the different annealing conditions, we performed curve fittings of C 1*s* core-level spectra in samples A, B, C, and D using Doniach–Sŭnjić curves convoluted with Gaussian distribution with 0.5 eV full width at half maximum [24] (Figure 4a). The background due to inelastic scattering was subtracted via the Shirley (integral) method [25]. Interestingly, we observed a very small defect intensity in the B sample. However, the C and D samples were still shown with the CH_3_NH_2_ molecular defect. From the curve-fitting data, we calculated the relative intensity area for each sample (Figure 4b). In the A sample, we confirmed over 18% of the concentration of the CH_3_NH_2_ molecular defect. However, the molecular defect had almost disappeared after annealing at 110 °C for 45 min, which is a typical annealing condition for a solar cell application. Interestingly, the CH_3_NH_2_ molecular defect still remained in the samples C and D annealed at 150 °C for 10 min and 30 min, respectively. 

The THz real conductivities of all samples show a strong resonance feature, which can be fitted by Lorentzian oscillator model (Equation (1)):(1)$$\tilde{\varepsilon }\left(\omega \right)=\varepsilon_{\infty }+\underset{j}{\sum }\frac{\Omega_{j}{}^{2}}{\left(\omega_{0j}{}^{2}-\omega^{2}\right)-i\gamma_{j}\omega }$$
where *ω_0j_*, *Ω_j_*, and *γ_j_,* are resonance frequency, oscillator strength, and the scattering rate of jth cosillator, respectively [23] (Figure 5a,b). The permittivity constant in the high frequency limit ε_∞_ is not in this experimental spectral range. The spectra are well fitted by three resonance peak frequencies of 1, 1.6, and 2 THz (Table 1). The two peaks around 1 and 2 THz are associated with the Pb–I vibrations of the inorganic components [13,26]. There is no correlation between the oscillator strength (*Ω*/2π) of these peaks and the CH_3_NH_2_ molecular defect. However, in the case of the peak at 1.6 THz, we can clearly observe the correlation between the oscillator strength and the relative intensity area of the CH_3_NH_2_ molecular defect (Figure 5c).

## 4. Conclusions

In our previous article, we found that the significant THz-wave absorption property originated in the CH_3_NH_2_ molecular defect-incorporated hybrid perovskite [13]. However, the relationship between the strength of THz-wave absorption and the CH_3_NH_2_ molecular defect was not understood quantitatively. It is important to control a THz-wave sensitivity for THz-based applications, such as sensors and modulators. To control the THz-wave oscillator strength in the range of 0.5 to 2.5 THz in a MAPbI_3_ hybrid perovskite thin film, in this article, we needed to control the density of the CH_3_NH_2_ molecular defect, which is controlled by a simple post-annealing process. There is the linear correlation between the defect density and the THz-wave oscillator strength at the 1.6 THz significant absorption in the MAPbI_3_ hybrid perovskite formed by the SVE method. For this to be a new application such as for THz-wave modulation, sensing, and imaging devices, we believe more wide-range and high absorptance properties are needed. For future research, additionally, we suggest an exchange or mixture of metal cation (Pb and Sn) and halogen anion (Cl, Br, and I) elements that will be induced with the different vibration modes in cation and anion structures of the hybrid perovskite structure.

## Figures and Tables

**Figure 1 nanomaterials-10-00721-f001:**
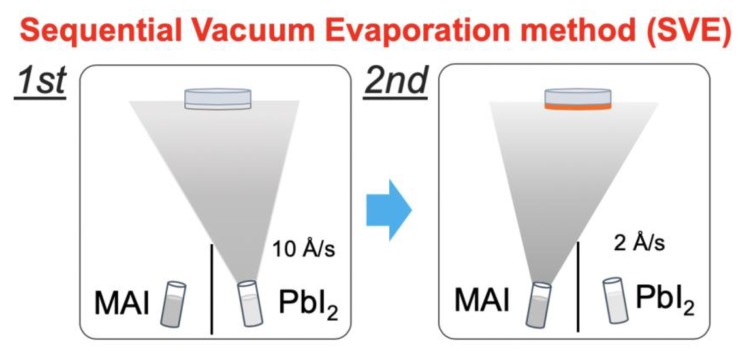
The schematic of the sequential vacuum evaporation (SVE) method.

**Figure 2 nanomaterials-10-00721-f002:**
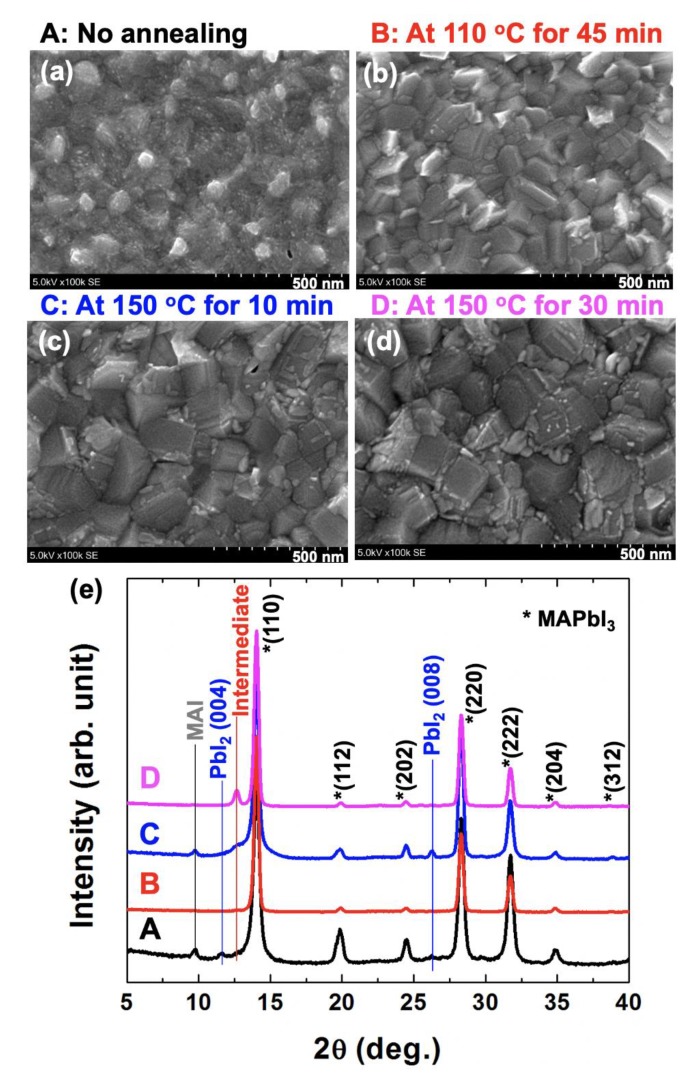
(**a**–**d**) The surface morphologies by SEM measurements and (**e**) XRD results with variable post-annealing conditions, A: No annealing, B: At 110 °C for 45 min, C: At 150 °C for 10 min, and D: At 150 °C for 30 min. In the typical annealing condition (B: for solar cell application) for MAPbI_3_ (*), MAI, PbI_2_, and intermediate phases were not observed.

**Figure 3 nanomaterials-10-00721-f003:**
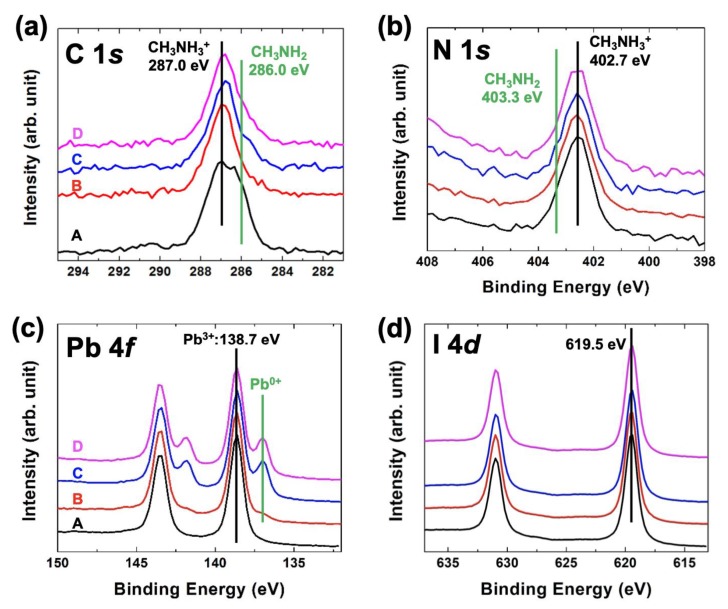
(**a**) C 1*s*, (**b**) N 1*s*, (**c**) Pb 4*f*, and (**d**) I 4*d* Core-level spectra with A: As received, B: 110 °C/45 min, C: 150 °C/10 min, and D: 150 °C/30 min in N_2_ atmosphere. We observed clearly the chemical states of (1) CH_3_NH_2_ molecular defect and (2) Pb^0+^ metal.

**Figure 4 nanomaterials-10-00721-f004:**
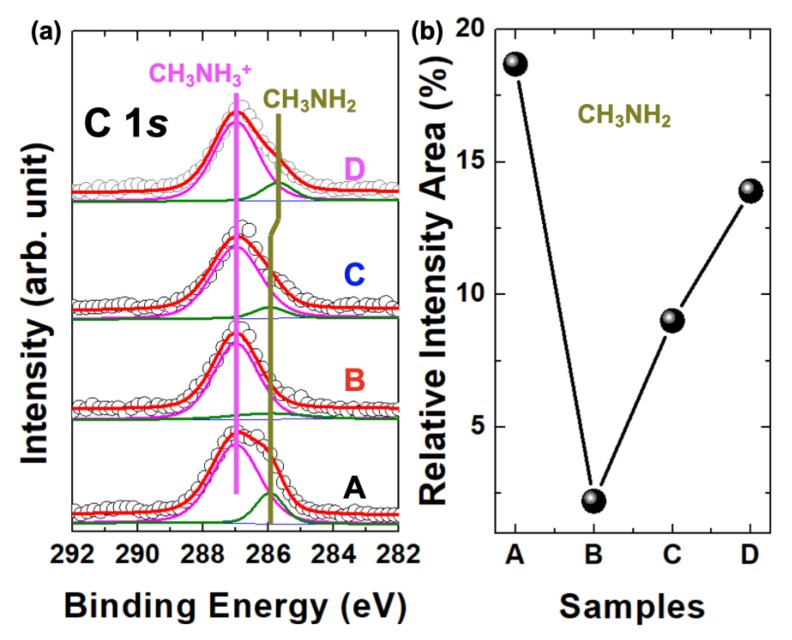
(**a**) Curve fittings for C 1*s* core-levels of A, B, C, and D. (**b**) The relative intensity areas for the chemical state of the CH_3_NH_2_ molecular defect.

**Figure 5 nanomaterials-10-00721-f005:**
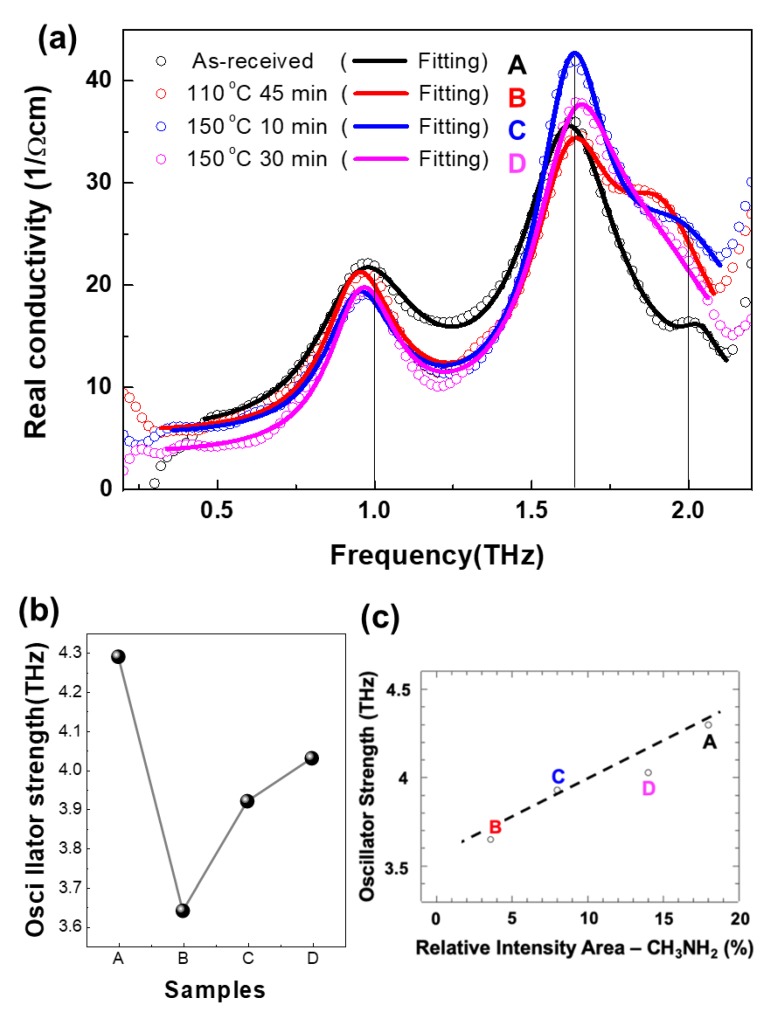
(**a**) THz real conductivities of A, B, C, and D. The Lorentz oscillator model fitting results are plotted by line. (**b**) The fitting results of oscillator strength of the 1.6 THz resonance peak. (**c**) The correlation between (**b**) and the relative intensity area for the chemical state of the CH_3_NH_2_ molecular defect (Figure 3b).

**Table 1 nanomaterials-10-00721-t001:** The fitting results of collection of Lorentz oscillators.

Sample	*ω_0_*/2π [THz]	*Ω*/2π [THz]	γ/2π [THz]
A	0.97	3.19	0.389
	1.62	4.29	0.37
	2.03	1.14	0.15
B	0.95	2.66	0.272
	1.62	3.64	0.319
	1.91	3.36	0.395
C	0.95	2.51	0.276
	1.63	3.92	0.274
	1.98	3.88	0.549
D	0.96	2.66	0.269
	1.64	4.03	0.343
	1.9	3.66	0.59

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
