# Peer review of "Strong Linear Correlation between CH3NH2 Molecular Defect and THz-Wave Absorption in CH3NH3PbI3 Hybrid Perovskite Thin Film"

_nanomaterials, 2020, doi:10.3390/nano10040721_

Round 1

Reviewer 1 Report

It seems that the overall direction and significance of this paper can be viewed positively. However, I think some contents should be supplemented and corrected before evaluating the quality of the contents of this paper.

Q1. What does SVE signify? (line 56)

Q2. It seems that 280nm of CH3NH3I is stacked on 100nm thick PbI2. How can the final thickness be 300nm? (line 71 ~ 74)

Q3. I think you must add ‘(XPS)’ behind ‘x-ray photoelectron spectroscopy’. (line 79)

Q4. “The XRD is RINT-TTR3/NM with CuK(alpha) source made by Rigaku (line 80 ~ 81)”, what does the sentence mean?

Q5. It seems that the experiment was conducted in the region between 0.2 and 2.2 THz. (line 87 ~ 88) Then, what is the basis for saying that you can control oscillator strength between 0.5 and 2.5 THz? (line 159)

Q6. It seems to be one of the key points of the paper is conducting experiments by changing the temperature from 110 to 150 C and the annealing time from 10 to 45 minutes.

1) What happens when you experiment below 110 C?

2) What happens if the temperature is over 150 C for a short time (within 10 minutes)?

3) Is it okay not to mention data at 120,130,140 C?

4) Is it okay not to mention data at 15, 30, and 45 minutes?

5) What happens if you heat the samples over 45 minutes?

If there is data to answer these questions, please show me data, and if not, I think it would be better for you to do additional experiments.

Q7. In Figure 1, compared with the photo in (b)-(d), the photo in (a)’s brightness seems different.

Q8. In (a)-(d) of Figure 1, the scale bar of the picture is not clearly visible. It seems to be necessary to make this look clearer.

Q9. no-annealing > no-annealed (line 97)

Q10. The annealed temperature > The annealing temperature (line 100)

Q11. What does the ‘and it looks a degradation’ in line 102 mean?

Q12. For Samples A and B in Figure 1, the peak at *(312) is not clearly visible.

Q13. In line 113, I think it is better to mention in the text that the temperature was kept constant at ‘As increasing the annealing time from 10 to 30 min’.

Q14. What does “The A sample was observed consistently with our previous reports.20” mean? (line 125)

Q15. I think it is necessary to write what the binding energy of the CH3NH3+ cation and the CH3NH2 molecular defect in the case of N 1s is also. (line 125 ~ 127)

Q16. annealing temperatures > annealing conditions (line 136)

Q17. In Figure 4(a),

1) What is the unit of Real conductivity in this text?

2) How about increasing the line thickness and reducing the size of the dots to increase visibility?

3) How about matching the color of the lines in the graph and the colors of the letters A, B, C, D?

Q18.

1) What is the name of Table1?

2) There is no interpretation about (gamma) / 2 * pi in the tex. Then, why did you put it in the table?

Q19. Finally, compared with ‘Maeng, I., Lee, Y.M., Park, J. et al. Significant THz absorption in CH3NH2 molecular defect-incorporated organic-inorganic hybrid perovskite thin film. Sci Rep 9, 5811 (2019).’ What is the most crucial difference between this article and that article? How about mentioning it in the conclusions?

Author Response

Dear Editor, (cc: Reviewer 1)

Thank you so much and the reviewer for the valuable comments and suggestions. We have considered them carefully and have made substantial changes accordingly. Below please find our point-by-point responses to the reviewer comments. Enclosed please also find the revised manuscript.

Sincerely,

Min-Cherl JUNG

Reviewer 2 Report

The manuscript from I. Maeng et al. reports on the relation between terahertz (THz) waves absorption and CH3NH2 molecular defect in methylammonium lead iodide perovskite (CH3NH3PbI3) films. The authors found a linear correlation between the molecular defect density and the oscillator strength at 1.6 Thz.

The paper is clearly written and sounds good. It deserves to be published in nanomaterials journal after considering corrections as follow:

  1. The authors have to define correctly all the abbreviations used for the first time for the comfort of the readers, for example SVE line 56, page 2.

  1. Page 4, line 125, add Ref. 20 with brackets.

  1. Page 6, line 157, Table 1 caption is missing.

  1. Values in Table 1 are not understandable and need further explanation. The fitting parameters and the relation with material properties have to be further explained and justified.

Author Response

Dear Editor, (cc: Reviewer 2)

Thank you so much and the reviewer for the valuable comments and suggestions. We have considered them carefully and have made substantial changes accordingly. Below please find our point-by-point responses to the reviewer comments. Enclosed please also find the revised manuscript.

Sincerely,

Min-Cherl JUNG

Round 2

Reviewer 1 Report

I am very pleased to see many improvements over the first version. In particular, I was able to understand a number of things I was curious about. However, there are still some questions for me.

<Grammar issues>

These sentences seem to have grammatical errors. Please check these sentences.

  1. (line 37) Currently, the number of published works on OHPs are growing exponentially.
  2. (line 50) However, another problem is being such as the material stability
  3. (line 68-70) A silicon substrate (n-type doped Si(100)) were cleaned by sonication in acetone for 10 min, rinsed in heated acetone for 1 min, and then UV-Ozone treatment for 30 min before loading into a vacuum chamber.
  4. (line 100) there is no MAI, PbI2 and intermediate phases.
  5. (line 160-161) The permittivity constant in the high frequency limit ?∞ is not in his experimental spectral range.

<Reader-friendly/Readability issues>

Please check these sentences for helping readers to read easily.

  1. (line 24-26) the thin film is confirmed without CH3NH3I and PbI2 with increasing the density of CH3NH2 molecular defect.
  2. (line 42) (At 1.5~1.7 THz with 20~50 % absorptance)
  3. The letter you sent me earlier helped me understand how the temperature and time conditions were set. However, I think that the description of the sentence needs to be reinforced by reflecting the contents of the letter that you sent me for future readers.

(line 80-81) Over the temperature of 150 °C, the depletion/degradation processes of MAPbI3 is so fast and very difficult to maintain a hybrid perovskite structure.

  1. (line 114-116) This result is similar to several reports and we confirmed that the typical annealing conditions can remove MAI and PbI2 phases formed by the SVE method.

<Other issues>

  1. (line 42) in MAPbI3 and FAPbI3 originated from a kind of defect structure. > How about writing full names to MAPbI3 and FAPbI3?
  2. Please review the verb tense throughout the paper. Currently, there is still little coherence.
  3. There have been some modifications, but the spectral range in each sentence still does not match. What is the reason?

(line 92 & 171) We measure the THz time 91 domain signal in the spectral range from 0.2 up to 2.5 THz. & To control the THz-wave oscillator 170 strength in the range of 0.5 to 2.5 THz in MAPbI3 hybrid perovskite thin film,

  1. In my opinion, there seems to be insufficient data to reach the conclusions mentioned below. In order to be sure of the following conclusions, I think that it is necessary to 1) increase the temperature at the same time condition and 2) increase the time at the same temperature condition.

(line 107-109) From these SEM results, we can agree that 1) the annealing process with the high temperature can increase the grain size and 2) the increased annealing time can make a degradation at the grain boundary.

  1. Is there data at 100 C? Please review the following sentence.

(line 116-117) When we increased the annealing temperature 116 from 100 to 150 °C (the sample C)

  1. The following details are not clearly visible in the graphs.

(line 130-132) and CH3NH2 molecular defect (C 1s - 286.0 eV and N 1s - 403.3 eV) are observed in the C 1s and N 1s core-level spectra. [13,21] (Fig. 3a and 3b) The peak intensity of CH3NH2 molecular defect was decreased after annealing,

  1. In my opinion, in the result section, a brief introduction is required before explaining the results of SEM experiments and terahertz experiments.

Regards,

Author Response

Dear Editor,

Thank you so much and the reviewer for the valuable comments and suggestions. We have considered them carefully and have made substantial changes accordingly. Below please find our point-by-point responses to the reviewer comments. Enclosed please also find the revised manuscript.

Sincerely,

Min-Cherl JUNG

Response to Reviewer #1

I am very pleased to see many improvements over the first version. In particular, I was able to understand a number of things I was curious about. However, there are still some questions for me.

<Grammar issues>

These sentences seem to have grammatical errors. Please check these sentences.

  1. (line 37) Currently, the number of published works on OHPs are growing exponentially.

 We have modified. “At the last decade, many works are reported in OHP material research”

  1. (line 50) However, another problem is being such as the material stability

 We have modified. “However, another problem is arising such as the material stability”

  1. (line 68-70) A silicon substrate (n-type doped Si(100)) were cleaned by sonication in acetone for 10 min, rinsed in heated acetone for 1 min, and then UV-Ozone treatment for 30 min before loading into a vacuum chamber.

 We have modified. “A silicon substrate (n-type doped Si(100)) were cleaned by sonication in acetone for 10 min and then it was rinsed in heated acetone for 1 min. Lastly, UV-Ozone treatment for 30 min before loading into a vacuum chamber.”

  1. (line 100) there is no MAI, PbI2 and intermediate phases.

 We have modified. “MAI, PbI2 and intermediate phases were not observed.”

  1. (line 160-161) The permittivity constant in the high frequency limit ?∞ is not in his experimental spectral range.

 We have added “t” in “his” (his  this).

<Reader-friendly/Readability issues>

Please check these sentences for helping readers to read easily.

  1. (line 24-26) the thin film is confirmed without CH3NH3I and PbI2 with increasing the density of CH3NH2 molecular defect.

 We have modified. “And CH3NH3I and PbI2 were disappeared in the thin film after the post-annealing at 150 °C for 30 min. However, the density of CH3NH2 molecular defect was increased.”

  1. (line 42) (At 1.5~1.7 THz with 20~50 % absorptance)

 We have removed it to avoid any confusion for readers.

  1. The letter you sent me earlier helped me understand how the temperature and time conditions were set. However, I think that the description of the sentence needs to be reinforced by reflecting the contents of the letter that you sent me for future readers.

(line 80-81) Over the temperature of 150 °C, the depletion/degradation processes of MAPbI3 is so fast and very difficult to maintain a hybrid perovskite structure.

 Thank you for your kind advise. We have modified in the Materials and Methods section. “To see the temperature dependence of MAPbI3, we performed the annealing processes at the temperature of 110 °C for 45 min (the typical annealing condition for solar-cell application) and 150 °C for 10 and 30 min. (to avoid any dramatical material degradation).[22]”

  1. (line 114-116) This result is similar to several reports and we confirmed that the typical annealing conditions can remove MAI and PbI2 phases formed by the SVE method.

 We have modified. “We confirmed that the typical annealing condition for solar-cell application could remove the MAI and PbI2 phases formed by the SVE method.”

<Other issues>

  1. (line 42) in MAPbI3 and FAPbI3 originated from a kind of defect structure. > How about writing full names to MAPbI3 and FAPbI3?

 Thank you for your suggestion. We have modified. “~CH3NH3PbI3 (MAPbI3) and HC(NH2)2PbI3 (FAPbI3)~”

  1. Please review the verb tense throughout the paper. Currently, there is still little coherence.

 We have performed. Thank you for your kind notice.

  1. There have been some modifications, but the spectral range in each sentence still does not match. What is the reason?

(line 92 & 171) We measure the THz time 91 domain signal in the spectral range from 0.2 up to 2.5 THz. & To control the THz-wave oscillator 170 strength in the range of 0.5 to 2.5 THz in MAPbI3 hybrid perovskite thin film,

 Thank you for your notice. As you recommended, we have fixed it with “from 0.5 to 2.5 THz” to avoid any confusion for readers.

  1. In my opinion, there seems to be insufficient data to reach the conclusions mentioned below. In order to be sure of the following conclusions, I think that it is necessary to 1) increase the temperature at the same time condition and 2) increase the time at the same temperature condition.

(line 107-109) From these SEM results, we can agree that 1) the annealing process with the high temperature can increase the grain size and 2) the increased annealing time can make a degradation at the grain boundary.

 We have added several references to confirm this explanation. [13,14,20,21]

  1. Is there data at 100 C? Please review the following sentence.

(line 116-117) When we increased the annealing temperature 116 from 100 to 150 °C (the sample C)

 Thank you so much for your kind notice again. We have fixed it.

  1. The following details are not clearly visible in the graphs.

(line 130-132) and CH3NH2 molecular defect (C 1s - 286.0 eV and N 1s - 403.3 eV) are observed in the C 1s and N 1s core-level spectra. [13,21] (Fig. 3a and 3b) The peak intensity of CH3NH2 molecular defect was decreased after annealing,

 It can make clear by the reference of 13 and 21.

  1. In my opinion, in the result section, a brief introduction is required before explaining the results of SEM experiments and terahertz experiments.

 We have added. “However, the relationship between the strength of THz-wave absorption and the CH3NH2 molecular defect was not understood quantitatively. It is important to control a THz-wave sensitivity for THz-based applications such as sensor and modulator.”

Additionally, the authors would like to thank deeply for the reviewer’s valuable comments.
